# From Chronic Lymphocytic Leukemia to Plasmablastic Myeloma: Beyond the Usual Richter Transformation

**DOI:** 10.3390/curroncol32100550

**Published:** 2025-09-30

**Authors:** Mathias Castonguay, Marie-France Gagnon, Alexandre Le Nguyen, Rafik Terra, Sarah-Jeanne Pilon, Guylaine Lépine, Richard LeBlanc, Jean Roy, Sandra Cohen, Isabelle Fleury, Luigina Mollica, Olivier Veilleux, Jean-Sébastien Claveau

**Affiliations:** 1Hematology-Oncology and Cell Therapy Institute, Hôpital Maisonneuve-Rosemont, Université de Montréal, Montréal, QC H1T 2M4, Canada; 2Faculty of Medicine, Université de Montréal, Montréal, QC H3C 3J7, Canada; 3Pathology and Cellular Biology Department, Maisonneuve-Rosemont Hospital, Université de Montréal, Montréal, QC H1T 2M4, Canada

**Keywords:** chronic lymphocytic leukemia, Richter transformation, myeloma, plasmablastic myeloma

## Abstract

Transformation of chronic lymphocytic leukemia (CLL) into a lymphoproliferative neoplasm with plasmablastic differentiation is exceptionally rare and remains poorly characterized. We report the first case of a patient with CLL developing a clonally related plasmablastic myeloma (PBM). The disease was defined by the emergence of a new del(17p) and TP53 mutation. Clinically, the PBM displayed an aggressive course, with primary refractoriness to frontline daratumumab, bortezomib, lenalidomide, and dexamethasone (Dara-VRd), leading to rapid fatality. This case highlights the importance of recognizing plasmablastic Richter transformation (RT) as a distinct entity in future classifications, given its unique biology and resistance profile compared with classical RT.

## 1. Introduction

Chronic lymphocytic leukemia/small lymphocytic lymphoma (CLL/SLL) is a lymphoproliferative syndrome characterized by small monomorphic CD19+ B-cells that frequently co-express CD5, CD23, and CD200, with weak CD20 and surface immunoglobulin expression [1]. The World Health Organization (WHO) classification of hematolymphoid tumors defines Richter transformation (RT) as the histologic transformation of CLL to either diffuse large B-cell lymphoma (DLBCL-RT) or Hodgkin lymphoma, and the vast majority (>80%) of RT are DLBCL [1,2]. DLBCL occurring in a patient with CLL may or may not have clonal relation to the preceding CLL, and clonally related DLBCL-RT tends to have different biology and worse prognosis than the de novo subtype [3].

Lymphoproliferative neoplasms with plasmablastic morphology mostly comprise plasmablastic lymphoma (PBL) and plasmablastic myeloma/plasmocytoma (PBM) [4]. PBL is a lymphoma comprised of plasmablast cells, activated B-cells that have undergone somatic hypermutation and class switching recombination and are transitioning into plasma cells. In both PBM and PBL, CD19 and CD20 are usually negative, whereas IRF4-MUM1, CD38, and CD138 are positive. PBL is frequently associated with HIV and/or EBV. *MYC* gene rearrangements occur in approximately 75% of EBV-positive PBL and 43% of EBV-negative PBL [5]. In contrast to PBL, PBM is not recognized as a distinct entity in the WHO 2022 classification but rather refers to a morphological variant of multiple myeloma (MM) with plasmablastic morphology. No specific criteria exist, but a cutoff of >2% plasmablastic cells on the bone marrow examination has been used [6]. Morphology and immunophenotype of PBM may be indistinguishable from PBL [4,7]. Extra-medullary involvement is frequent in PBM, with high rates of soft tissue/skin, nodal, pleural, and lung involvements being reported [8]. Distinguishing PBM from PBL can be challenging, although EBV-positivity may help as PBM is EBV-negative [7]. Moreover, no association between HIV or HHV8 has been described with PBM. Light chain restriction and cyclin D1 expression are more common in PBM. Typical genetic features of MM favor PBM over PBL, although both entities are associated with *MYC* rearrangements [1]. Whether PBM is associated with high-risk MM cytogenetic abnormalities (e.g., del(17p), t(4;14), t(14;16)) remains unclear, as several reports have shown conflicting associations [9,10]. Typical MM-defining events (hypercalcemia, lytic bone lesions, renal impairment, and anemia), along with the absence of significant lymphadenopathy might also help to distinguish PBM from PBL [7,11]. Currently, there are no established treatment guidelines for PBM; however, management typically involves aggressive anti-MM therapies such as Dara-VRd, Isa-KRd, or PACE. Historically, PBM has been associated with a shorter survival, but recent studies rarely report plasmablastic morphology, limiting contemporary prognostic insight.

## 2. Detailed Case Presentation

We present herein the case of a patient with CLL evolving into PBM. The patient consented to this case report.

The patient was a 62-year-old man diagnosed with CLL in 2011, Rai Stage 1. FISH analyses with *TP53*/*CEP17* and *ATM*/*CEP11* probes were performed and revealed a deletion involving 11q/*ATM* in 96% of interphase nuclei. Initial therapy consisted of fludarabine, cyclophosphamide, and rituximab ending in February 2012 and achieved a complete response as defined by the iwCLL response assessment [12]. The patient relapsed in 2016 with progressive lymphocytosis, but retreatment was not indicated.

In August 2018, the patient developed significant night sweats and lower gastrointestinal bleeding. At that time, the lymphocyte count was 130 × 10^9^/L, hemoglobin 123 g/L, and platelets 129 × 10^9^/L. FISH results revealed persistence of the deletion involving 11q with no 17p deletion. *IGH* clonality was confirmed using a PCR-based approach, in accordance with EuroClonality/BIOMED-2 guidelines [13]. The immunoglobulin heavy chain variable region (*IGHV*) was unmutated, as assessed by an NGS-based approach (LymphoTrack Assay Panel, Invivoscribe, San Diego, CA, USA) [14]. A colonoscopy revealed a 5 mm sessile polyp, and pathological analysis was consistent with a CLL infiltrate. Scans and PET-CT showed multiple non FDG-avid and non-bulky lymphadenopathies in the cervical, thoracic and abdomen regions. Ibrutinib was started in October 2018, and the patient achieved a complete response. Ibrutinib was switched to Zanubrutinib after the patient experienced atrial fibrillation in April 2023.

In February 2024, the patient developed new thrombocytopenia (48 × 10^9^/L) associated with hemoptysis, epistaxis, and diffuse bone pain. LDH increased from 192 U/L (upper limit normal: 210) in November 2023 to 1499 U/L. Serum protein electrophoresis revealed a newly emerging IgA-Lambda monoclonal protein at 2.2 g/L. Serum free light chain assay revealed a Lambda and Kappa light chain concentrations of 3445.4 mg/L and 7.3 mg/L, respectively. Beta-2-microglobulin level was 5.58 mg/L. Hemoglobin, white blood cell counts, and liver functions were otherwise normal. The patient rapidly developed severe acute renal failure, presumably caused by cast nephropathy (renal biopsy was not performed), and ultimately required hemodialysis. PET-CT showed a new hypermetabolic lesion of 4 cm with a standardized uptake value (SUV) of 7.5 in the spleen, new small lymphadenopathies in the retro-pancreatic region with an SUV of 7.8, and para-aortic region with an SUV of 6.7, along with diffuse hypermetabolism of the bone marrow with an SUV of 6.4. Bone marrow aspirate and biopsy revealed massive infiltration (>95%) by mature and immature plasma cells. Flow cytometry analysis was remarkable for a 70% infiltration of cytoplasmic Lambda-restricted plasma cells, exhibiting the following immunophenotypic aberrations: CD45−/dim, CD19−, CD20−, CD28+, CD38++, CD138+/−, CD117+/−, CD81+/−, and CD27+/dim, CD56−, with concomitant 2% infiltration of classical CLL lymphocytes (Figure 1). The marrow biopsy revealed widespread infiltration by plasmablasts, with occasional mature plasma cells (Figure 2). These cells were positive for CD10, CD138, CD79a, MUM1, p53, and lambda light chain and negative for BCL-1, BCL-2, BCL-6, CD19, CD20, CD23, CD3, CD30, c-MYC, TdT, CD5, CD34, and CD56. The Ki67 proliferation index was 100%. In situ hybridization with EBER was negative. Next-generation sequencing (NGS) panel testing was performed at an outside reference laboratory and revealed a deleterious *TP53* mutation (NM_000546.4:c.476C>T p.(Ala159Val), VAF 43%) but was otherwise negative for other mutations. A FISH probe panel including *IGH*/*CCND1* (Abbott Molecular, Des Plaines, IL, USA), *IGH*/*FGFR3* (Abbott Molecular, Des Plaines, IL, USA), *IGH*/*MAF* (Abbott Molecular), and *IGH*/*MAFB* (CytoCell, Cambridge, UK) dual color dual fusion probe sets and 1p/1q (CytoCell) and *TP53*/*CEP17* (CytoCell) probe sets was performed and showed a 17p13.3(*TP53*) deletion in 97% of interphase nuclei. While no IGH fusion with *CCND1*, *FGFR3* (*NSD2*), *MAF*, or *MAFB* fusion was found, three signals at the IGH locus were detected, which could suggest a possible IGH rearrangement with another untested fusion partner. In addition, no deletion of 1p or gain of 1q was detected. Clonal relatedness of the CLL and plasmablastic neoplasm was confirmed by demonstrating a shared immunoglobulin gene rearrangement (*IGHV 3-33*01/IGHJ 5*02*) using an NGS-based approach (Invivoscribe, San Diego, CA, USA).

With a diagnosis of CLL evolving into PBM, a daratumumab, lenalidomide, bortezomib, and dexamethasone (Dara-VRd) regimen was initiated. The patient was primary refractory to Dara-VRd, with serum lambda free light chain levels progressing to 4571 mg/L. PET-CT scan performed on day 30 of Dara-VRd revealed progression of previously described splenic lesion and new abdominal lymphadenopathies. Transition to conventional intensive chemotherapy (such as PACE) was planned; however, the patient declined further treatment and succumbed to the disease two weeks later.

## 3. Discussion

To the best of our knowledge, there is no previous report describing a clonal evolution of CLL to PBM. Patients with CLL may be more at risk of developing MM, although the occurrence of both diseases remains rare and incidence unclear [15]. In a series of 28 patients with both CLL and MM, Dholoria et al. reported no difference in overall survival (OS) between patients with CLL and MM compared to MM patients. Moreover, no clonal relationship could be established [15].

Patients with CLL are at increased risk for both solid tumors and secondary malignancies, primarily due to defects in humoral and cellular immunity inherent to CLL, as well as treatment-related immunosuppression [16,17]. BTK inhibitors have been shown to modulate immune function through off-target effects on other kinases, such as interleukin-2–inducible T-cell kinase, and are particularly associated with an increased risk of non-melanoma skin cancers and solid tumors, although this risk appears to be lower than that observed with traditional chemoimmunotherapy regimens [18]. However, BTK inhibitors are not associated with an increased risk of RT. The risk factors for RT development are mainly disease-driven, with unmutated *IGHV* status, *TP53* mutations, *NOTCH1* mutations, and elevated baseline LDH levels being particularly associated with a higher risk of transformation [19,20].

The emergence of PBL in a patient with CLL is extremely rare, with no more than 15 cases reported to date [21]. Concomitant diagnosis of CLL and PBL, and PBL arising from pre-treated CLL (including with Ibrutinib) have been described. Most PBLs arising in CLL patients were EBV-negative, in contrast to de novo PBL. Acquisition of *MYC* rearrangement and P53 expression or *TP53* mutation were frequent. Most patients died shortly after diagnosis due to disease refractoriness or progression after a short-lived response. Assessing the clonal relationship between the plasmablastic neoplasm and the underlying CLL is important, as CLL-associated immunosuppression could represent a risk factor for the development of de novo PBL/PBM rather than truly clonally related PBL/PBM RTs.

In our case, the presumed diagnosis of CLL evolving into PBM is based on the massive bone marrow infiltration by mature EBV negative plasma cells with plasmablasts, in the absence of immunoblastic differentiation. Although no lytic bone lesions were detected, the presence of a monoclonal IgA protein, a markedly elevated serum lambda free light chain level, and a presumed cast nephropathy favored PBM over PBL. The patient was refractory to frontline treatment despite using an effective and modern anti-myeloma regimen (Dara-VRd).

## 4. Conclusions

CLL transforming into a lymphoproliferative neoplasm with plasmablastic differentiation (PBL or PBM) is a rare and catastrophic complication. Evolution is notably aggressive, characterized by high treatment refractoriness, with most patients succumbing within weeks to months following diagnosis. Hematologists should be aware of this particularly aggressive transformation of CLL, and larger cohort studies are needed to further characterize plasmablastic RTs, incorporate them into new classifications, and guide more effective treatment strategies.

## Figures and Tables

**Figure 1 curroncol-32-00550-f001:**
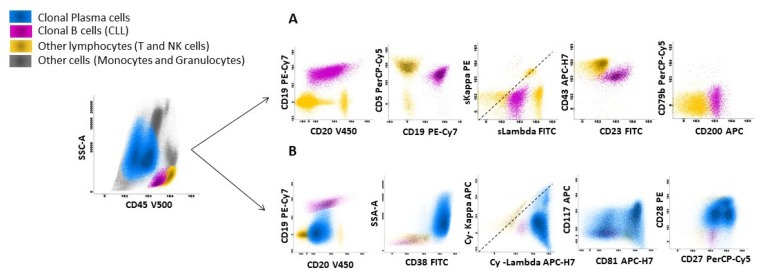
Flow cytometry analysis of clonal plasma cells with concomitant CLL population in bone marrow aspiration. Bone marrow infiltration by clonal plasma cells concomitantly with a CLL population: flow cytometry analysis was performed with FACSCanto-II and data analyzed using Infinicyt software (version 2.0.5). (**A**) CLL population (2%) (purple color) with a typical immunophenotype: CD19+ CD20+dim surface Lambda restriction +dim CD5+ CD23+ CD43+ CD200+ CD79b−. (**B**) Plasma cell clone (70%) (blue color) showing cytoplasmic lambda restriction with the following aberrations: CD45-/dim CD19− CD20− CD38++ CD117+/− CD81+/− CD27+/dim CD28+.

**Figure 2 curroncol-32-00550-f002:**
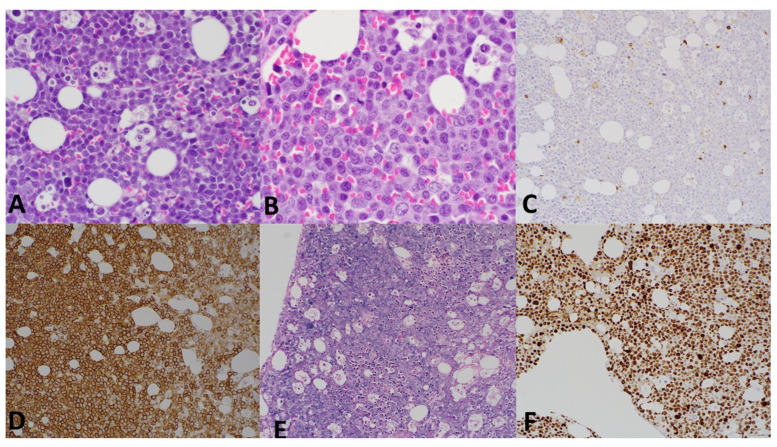
Immunohistochemistry and corresponding H&E image of bone marrow infiltration by plasmablastic proliferation. (**A**) H&E, 400× magnification. (**B**) H&E, 600× magnification. (**C**) CD5 immunohistochemistry, 400× magnification. (**D**) CD138 immunohistochemistry, 400× magnification. (**E**) In situ hybridization with lambda probe, 400× magnification. (**F**) Ki67 immunohistochemistry, 400× magnification.

## Data Availability

Request regarding data should be directed to the corresponding author.

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
