# Peer review of "From Chronic Lymphocytic Leukemia to Plasmablastic Myeloma: Beyond the Usual Richter Transformation"

_curroncol, 2025, doi:10.3390/curroncol32100550_

Round 1
Reviewer 1 Report
Comments and Suggestions for Authors
1: Include a paragraph or 2 to outline why secondary malignancies are likely in patients with CLL/SLL as part of discussion
2: include a paragraph on secondary malignancies associated with BTK inhibitors as part of discussion.
3: IgA associated myelomas are usually poorly differentiated and/or aggressive, as observed in this patient. Could you show a graphic evidence of clonal relationship between the CLL and the myeloma?
4: Could this patient have had a prior MGUS before the CLL?
Author Response
Reviewer 1
- Include a paragraph or 2 to outline why secondary malignancies are likely in patients with CLL/SLL as part of discussion
Response: A paragraph was added in the discussion (lines 184-194, 212-215).
- Include a paragraph on secondary malignancies associated with BTK inhibitors as part of discussion.
Response: A paragraph was added in the discussion (lines 186-190).
- IgA associated myelomas are usually poorly differentiated and/or aggressive, as observed in this patient. Could you show a graphic evidence of clonal relationship between the CLL and the myeloma?
Response: I apologize, but we were unable to provide graphical evidence of the clonal relationship. Clonality was demonstrated by identifying the same IGHV V and J sequences in both the CLL and PBM samples (now specified at line 148).
- Could this patient have had a prior MGUS before the CLL?
Response: No serum electrophoresis or light chain essay was performed prior to transformation unfortunately.
Reviewer 2 Report
Comments and Suggestions for Authors
- The authors report the first case of CLL evolving into plasmablastic myeloma (PBM). If so, I’s odd to conclude (in the Conclusions section) “ ….., with most patients succumbing within …” since this is the only case.
- This is a well-illustrated case. However, a single case does not warrant a new classification of plasmablastic RT as a distinct entity. More similar cases are definitely needed. The wording should be revised.
- Please italicize all the names of genes in the text.
- For Figure 2, a high-power cytology showing the plasmablastic cells might be more useful that Panels 2B (CD20-negative) and 2C (CD5-negative). If the authors prefer to keep Panels of CD5 and CD20, please add additional panel(s) of the marrow cytology. Figure 2E legends. The CD138 seems to be redundant.
- The author might cite the following review paper detailing the differential diagnosis lymphoid neoplasms with plasmablastic differentiation (plasmablastic lymphoma vs. plasmablastic myeloma). PMID: 31725418.
Author Response
Reviewer 2
- The authors report the first case of CLL evolving into plasmablastic myeloma (PBM). If so, I’s odd to conclude (in the Conclusions section) “ ….., with most patients succumbing within …” since this is the only case.
Response: We apologize for the misunderstanding. We were referring to lymphoproliferative neoplasms with plasmablastic morphology (namely plasmablastic lymphoma and plasmablastic myeloma). We have added this clarification in the revised manuscript. (line 225)
- This is a well-illustrated case. However, a single case does not warrant a new classification of plasmablastic RT as a distinct entity. More similar cases are definitely needed. The wording should be revised.
Response: We respectfully disagree with the reviewer. We believe that the documentation of one case of PBM and 15 cases of PBL supports considering plasmablastic lymphoproliferative disorders (PBM or PBL) as a distinct RT entity. Such a classification would facilitate better disease characterization and contribute to the development of improved treatment strategies.
- Please italicize all the names of genes in the text.
Response: Gene names are now italicized in the revised version of the manuscript.
- For Figure 2, a high-power cytology showing the plasmablastic cells might be more useful that Panels 2B (CD20-negative) and 2C (CD5-negative). If the authors prefer to keep Panels of CD5 and CD20, please add additional panel(s) of the marrow cytology. Figure 2E legends. The CD138 seems to be redundant.
Response: Figure 2 was updated accordingly.
- The author might cite the following review paper detailing the differential diagnosis lymphoid neoplasms with plasmablastic differentiation (plasmablastic lymphoma vs. plasmablastic myeloma). PMID: 31725418.
Response: We added the reference in the revised version.